# Cathodoluminescence Spectroscopic Stress Analysis for Silicon Oxide Film and Its Damage Evaluation

**DOI:** 10.3390/ma13204490

**Published:** 2020-10-10

**Authors:** Shingo Kammachi, Yoshiharu Goshima, Nobutaka Goami, Naoaki Yamashita, Shigeru Kakinuma, Kentaro Nishikata, Nobuyuki Naka, Shozo Inoue, Takahiro Namazu

**Affiliations:** 1Department of Mechanical and Electrical System Engineering, Faculty of Engineering, Kyoto University of Advanced Science, 18 Gotanda-cho, Yamanouchi, Ukyo-ku, Kyoto 615-8577, Japan; 2020mm02@kuas.ac.jp; 2R&D Department, HORIBA, Ltd., Miyanohigashi, Kisshoin, Minami-ku, Kyoto 601-8510, Japan; yoshiharu.goshima@horiba.com (Y.G.); shigeru.kakinuma@horiba.com (S.K.); kentaro.nishikata@horiba.com (K.N.); nobuyuki.naka@horiba.com (N.N.); 3Department of Mechanical Engineering, Faculty of Engineering, University of Hyogo, 2167 Shosha, Himeji, Hyogo 671-2201, Japan; inoue@eng.u-hyogo.ac.jp

**Keywords:** cathodoluminescence spectroscopy, SiO_2_ film, nondestructive stress analysis, Si nanocrystal, electron beam irradiation, Raman spectroscopy

## Abstract

We describe the stress analysis of silicon oxide (SiO_2_) thin film using cathodoluminescence (CL) spectroscopy and discuss its availability in this paper. To directly measure the CL spectra of the film under uniaxial tensile stresses, specially developed uniaxial tensile test equipment is used in a scanning electron microscope (SEM) equipped with a CL system. As tensile stress increases, the peak position and intensity proportionally increase. This indicates that CL spectroscopy is available as a stress measurement tool for SiO_2_ film. However, the electron beam (EB) irradiation time influences the intensity and full width at half maximum (FWHM), which implies that some damage originating from EB irradiation accumulates in the film. The analyses using Raman spectroscopy and transmission electron microscopy (TEM) demonstrate that EB irradiation for stress measurement with CL induces the formation of silicon (Si) nanocrystals into SiO_2_ film, indicating that CL stress analysis of the film is not nondestructive, but destructive inspection.

## 1. Introduction

Developing an experimental technique for non-contact and non-destructive stress distribution analysis of semiconductor devices and microelectromechanical systems (MEMS) is required for screening to eliminate imperfect products and for remaining life assessment of the products, which will lead to improving the reliability of the devices [1,2]. Raman spectroscopy is well-known as a powerful tool for measuring the surface stress of silicon (Si) with special resolution less than 1 μm without physical contact and damage [3,4,5]. Raman spectrum is sensitive to stress, so it is possible to draw a stress distribution map on a Si structure in a device. However, Raman analysis cannot be applied to wide-bandgap materials such as Si dioxide (SiO_2_). This is because the energy required when the electrons in the material are excited by the excitation light is large, and the detected Raman scattering light becomes weak. SiO_2_ film is, needless to say, a common material used as a passivation layer in electronics and Si MEMS devices [6]. In particular, Si MEMS devices possess movable or deformable mechanical elements composed of Si and SiO_2_ film; therefore, analyzing the stress of the SiO_2_ film as well as Si directly leads to developing mechanically reliable Si MEMS devices. Cathodoluminescence (CL) spectroscopy, based on a CL emission from a specimen by electron beam (EB) irradiation, is typically used to obtain information on the impurity [7] and dislocation [8] in ceramics. Recently, several researchers have used CL as a stress analysis tool because they have found that some of the CL spectral parameters are sensitive to stress in a specimen [9,10,11]. If CL spectroscopy is a non-destructive method for stress analysis, it will become a strong candidate as a tool for stress analysis of ceramics and oxides.

The objectives of this study are to apply CL spectroscopy to a quantitative stress analysis tool for a SiO_2_ film and to investigate sample damage originating from EB irradiation during stress analysis. An in-house uniaxial tensile test equipment is developed for stress application to the SiO_2_ film specimen to quantitatively investigate the relationships between the applied tensile stress and CL spectra from the film. Raman spectroscopy and transmission electron microscopy (TEM) are used to observe some structural changes originating from EB irradiation.

## 2. Experimental Procedure

Figure 1 illustrates a tensile test system operating in an SEM along with a photograph of tensile test equipment. A CL analysis system (HORIBA, Ltd. MP-32M, Kyoto, Japan) was equipped to a field-emission-type SEM (Hitachi High-Technologies, S-4300SE, Tokyo, Japan). A tensile stress applied to a film specimen can be controlled from the outside of the SEM chamber because electrical wirings between the equipment and a computer are connected automatically via a micro-connector once it is installed in the SEM. The tensile test equipment, which was designed to be of a compact size, consists of a piezoelectric actuator (PI Japan, P-840.95, Tokyo, Japan) in an aluminum (Al) alloy case, a load cell (Tech Gihan, TGRV08-2N, Kyoto, Japan), a specimen holder, and a micro-connector (Omnetics Connector Corporation, A28000-015, Minneapolis, MN, USA) [12,13,14,15]. The Al case has a lever structure, which can amplify the displacement of the actuator by a factor of 2.27 in the tensile direction. The applicable tensile elongation is 67 μm at the maximum. The force measurement resolution of the load cell is 2 mN (0.1 % of full scale, 2 N). The elongation of a specimen is measured by means of digital image processing using an SEM movie, of which the measurement resolution is approximately 13 nm/pixel after sub-pixel processing.

Figure 2 depicts a representative SiO_2_ film specimen for tensile testing by means of conventional MEMS fabrication techniques including photolithography, dry/wet etching, and thermal oxidation. The specimen is composed of a parallel gauge section including 200-nm-deep square concaves, four springs, two square chucking holes, and a frame [12,13,14]. An 820-nm-thick SiO_2_ film was coated on the entire surface of a 4-μm-thick single crystal Si specimen by wet thermal oxidation at 1100 °C. To check the tensile test setup for thin film stress evaluation, we first conducted a quasi-static uniaxial tensile test of a Si film specimen without the SiO_2_ film layer. The stress-strain relation was almost linear until brittle failure, although the graph is omitted here. Young’s modulus was calculated to be 167 GPa, which is almost the same as the bulk value, 168.9 GPa, for Si (110) [16]. This indicates that the test system has potential as stress application equipment for a thin film specimen.

CL light emitted from a SiO_2_ specimen by EB irradiation is collected by an ellipsoidal mirror and led by an optical fiber to a spectrometer with a charge coupled device (CCD) detector. The excitation voltage, probe current, and measurement time were set to 3 kV, 140 pA, and 5 s, respectively, for stress analysis. Obtained CL spectra under various uniaxial tensile stresses were fitted using the Gauss/Lorentz function to obtain CL spectral parameters.

## 3. Results and Discussion

Figure 3 shows a representative CL spectrum of a SiO_2_ film without external tensile stress. The spectrum is composed of three peaks related to different types of defects in SiO_2_. The red spectrum is attributed to the [≡Si-O-] band of the non-bridged oxygen hole center (NBOHC), the blue one is from the [≡Si] band of the trivalent Si, and the green one is from the [≡Si≡] band of the neutral oxygen vacancy defect. In this study, for stress analysis, the 1.85 eV peak related to NBOHC was used due to strongest peak among them [17].

Figure 4 shows the relationships between applied tensile stress and CL spectral parameters obtained from Gauss/Lorentz curve fitting. In all the experiments, EB was irradiated onto the flat surface of a SiO_2_ film specimen. In the experiment, the EB irradiation spot was unchanged, although the external tensile stress applied was changed in incremental steps. In Figure 4a, the peak position is found to be 1.8739 eV without external stress. With increasing external tensile stress, the peak position gradually increases. When a tensile stress of 280 MPa was applied, the peak position linearly increased to 1.8760 eV. By using a least-square method for data fitting, the increment rate was calculated to be 7.53 × 10^−3^ eV/GPa. In Figure 4b, the peak intensity also shows a trend similar to the peak position; that is, the intensity linearly increases with increasing tensile stress. By contrast, as shown in Figure 4c, the FWHM did not show a clear correlation with tensile stress because the bandwidth data changed randomly despite a monotonic change in the stress. Those CL analyses described above were conducted five times under the same condition; consequently, the trends obtained were consistent. Therefore, we could conclude that the peak position and intensity of the CL spectrum were sensitive to the external tensile stress applied.

Since the EB irradiation spot was unchanged irrespective of the tensile stress, the EB irradiation time accumulation as well as the tensile stress might have influenced the change in the CL spectral parameters. Therefore, we investigated the relationships between EB irradiation time and the parameters, as shown in Figure 5. In Figure 5a, the peak position at 5 s EB irradiation was 1.8692 eV. Although the irradiation time increased, the energy value at the peak position did not change that much. In the irradiation time range from 5 to 30 s, the difference between the maximum and minimum values was only 0.0007 eV. This fact suggests that the peak position was not influenced by EB irradiation. However, two other parameters clearly showed the irradiation time influence. The peak intensity, in Figure 5b, linearly decreases with increasing irradiation time, whereas the FWHM, in Figure 5c, linearly increases with the time.

From the investigation shown in Figure 4 and Figure 5, the peak position of the CL spectrum is a strong candidate as the representative parameter for SiO_2_ stress analysis. This is because the parameter was sensitive to only tensile stress, and it was insensitive to the EB irradiation time. The intensity and FWHM were obviously sensitive to the irradiation time. This experimental fact implies that some physical damage was introduced into the SiO_2_ film by EB irradiation accumulation. To investigate how the damage was introduced, Raman spectroscopy and TEM observations around EB irradiation points were conducted. As shown in Figure 6, EB spots were irradiated using an SEM onto Si and SiO_2_ film surfaces at 15 kV for 120, 180, 240, and 300 s. Each spot interval was set to be 10 μm. As shown in the SEM image, the irradiated spots formed on the SiO_2_ film presented a mottled pattern due to contamination or some damage.

Figure 7 shows representative Raman spectroscopic line scanning results showing (a) the Raman peak shift, (b) peak intensity, and (c) FWHM. The analysis was conducted using a commercial Raman spectroscope (HORIBA Jobin Yvon, LabRAM HR-800, Kyoto, Japan). The sample stage of the microscope was driven using a piezoelectric actuator in 1 μm steps. The black and gray plots are indicative of the analysis result on the SiO_2_ and Si surfaces, respectively. In Figure 7a, the peak position of each Raman spectrum obtained at each position on the Si surface remained roughly constant, ranging from 518.96 to 519.10 cm^−1^, which are close to the typical value for Si, 520 cm^−1^. However, in the case of the SiO_2_ surface, a different trend was obtained. For example, for the EB irradiated point for 120 s, the peak position shifted by around 0.1 cm^−1^ compared with the non-irradiated point. The amount of the peak shift monotonically increased to around 0.2 cm^−1^ with an increase in the irradiation time up to 300 s. Raman peak shift is well known to be related to stress applied to an analyzed point. The peak shift to the high cm^−1^ direction indicates that compressive stress was applied. As shown in Figure 7b,c, no wavy distribution occurred in the peak intensity and FWHM on both SiO_2_ and Si surfaces, indicating that there was no influence of EB irradiation on these parameters. Since the Raman peak shift is strongly influenced by the crystal structure of a material [18], the phenomenon, indicating Raman stress change around EB irradiation spots, was possibly caused by a structural change inside the SiO_2_ film or some kind of damage at the SiO_2_/Si interface caused by the irradiation. Based on the above results, however, no possibility of some SiO_2_/Si interface damage is expected.

Figure 8 shows TEM images of SiO_2_ thin films after EB irradiation. All specimens for TEM observation were prepared using an ion slicer (JEOL Ltd., EM-09100IS, Tokyo, Japan). After the preparation, the surface was irradiated with EB under the following conditions: 3 kV, 140 pA, and 5 s. For the observation, a TEM (FEI Company, Tecnai 20 ST, Hillsboro, OR, USA) was used, and the excitation voltage and emission current were set to 200 kV and 5.48 μA, respectively. Before EB irradiation, a typical amorphous structure was observed, although some low-contrast small dots were detected. After 5 s irradiation, several tens of dots shown as black mottles were clearly observed, with a diameter of around 2 nm. In spite of the very short-duration irradiation, those small dots were produced in the SiO_2_ film. After irradiation for 1000 s, numerous dots with approximately 3~5 nm diameter were obtained in the entire TEM image, which indicates that the quantum dots grew by successive EB irradiation. In the magnified photograph, the dot consists of a lattice pattern with approximately 0.2 nm intervals, which is in close agreement with the lattice spacing in the Si (220) plane [19]. This strongly suggests that EB-irradiation-induced nano-dots are probably nanocrystals made of single crystal Si. The generation of Si nanocrystals in SiO_2_ would be caused by Si-O network disconnection and oxygen desorption [20,21,22]. Because the nanocrystals were stressed by the SiO_2_ film, Raman stress distribution was probably changed around EB irradiation spots [23]. Also, a change in the CL spectrum was influenced by not only the SiO_2_ film structure change but also stress from SiO_2_ caused to Si nanocrystals. However, the mechanisms of changing Raman stress after EB irradiation, shown in Figure 7a, and changing CL intensity and FWHM shown in Figure 5b,c), should not be concluded until further experiments have been performed. Even though the EB irradiation time was just 5 s, which is the same irradiation energy (1.8 × 10^−4^ J/m^2^) as that in the CL stress analysis experiment, Si nanocrystals were generated. To measure the stress in SiO_2_ film using CL spectroscopy without any damage, an EB irradiation condition without Si nanocrystal formation must be specified.

## 4. Conclusions

We investigated the relationships between CL spectra of SiO_2_ film and external tensile stresses applied, for non-destructive stress analysis with a CL spectroscope. It was possible to estimate the stress on the SiO_2_ film directly from the CL peak shift because the peak position of the CL spectra linearly increased with increasing applied stress at a constant rate of 7.53 × 10^−3^ eV/GPa. However, Raman spectroscopy and TEM analysis suggested that Si nanocrystals were formed at the EB irradiation point on the SiO_2_ film, which indicated that the SiO_2_ film suffered from damage by EB irradiation. The Si nanocrystals generated affected the CL intensity and FWHM. For current CL stress analysis of SiO_2_ film, CL measurement for the shortest time would be necessary to limit damage.

In future, for realization of non-destructive CL stress analysis, it is required to quantitatively and experimentally understand the amount of Si nanocrystal formation in SiO_2_ film caused by EB irradiation, and then to establish the CL analysis condition to avoid the formation of Si nanocrystals. Moreover, considering CL stress analysis practical use in semiconductor industry, it is also important to consider the possibility of changes in CL spectra with elevated temperature.

## Figures and Tables

**Figure 1 materials-13-04490-f001:**
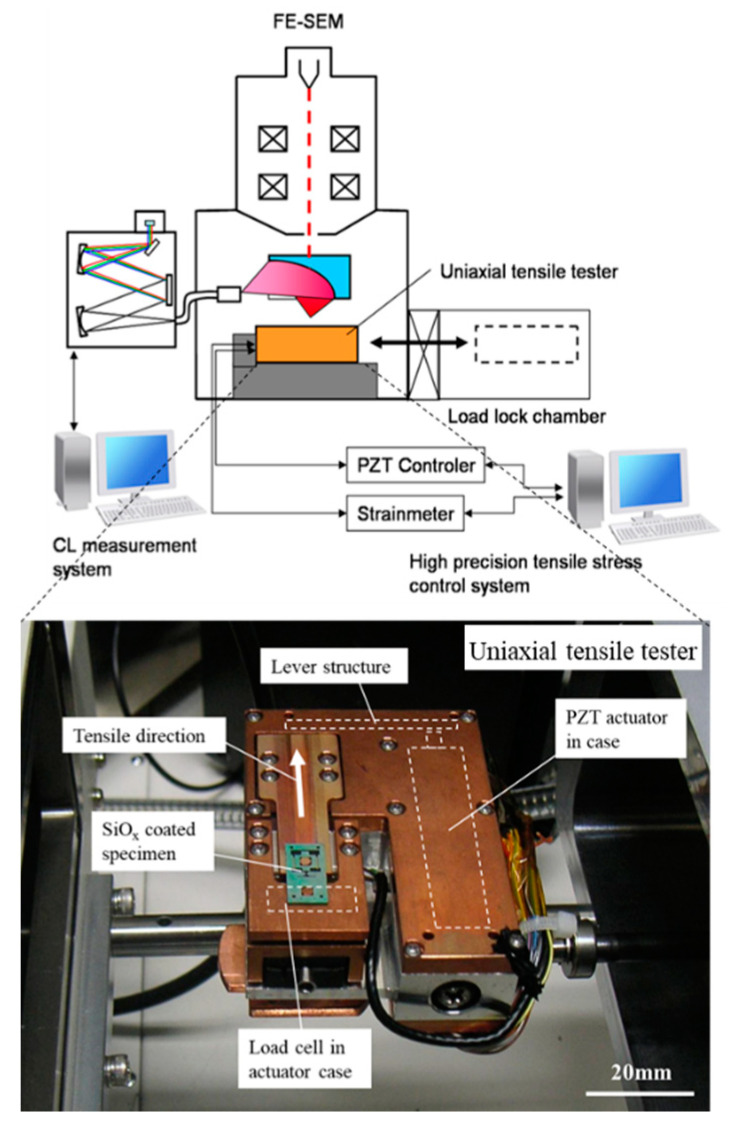
Schematic of the uniaxial tensile testing system along with the photograph of the uniaxial tensile tester.

**Figure 2 materials-13-04490-f002:**
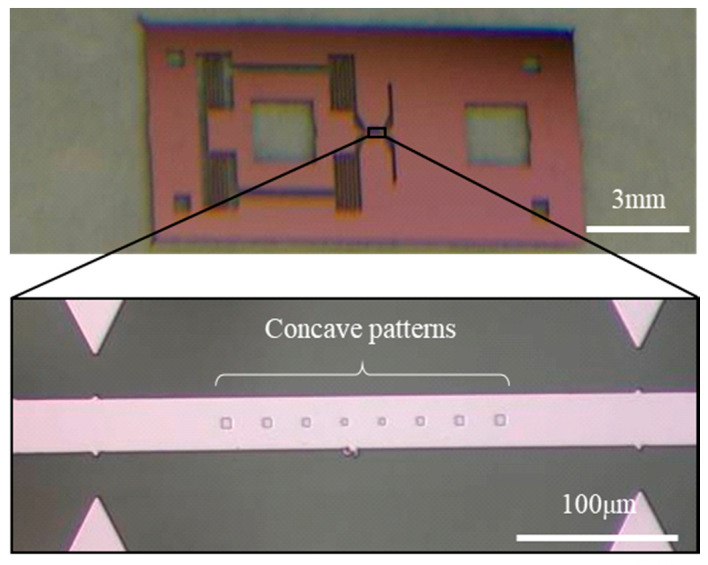
Photograph of a SiO_2_ film specimen for the tensile loading test.

**Figure 3 materials-13-04490-f003:**
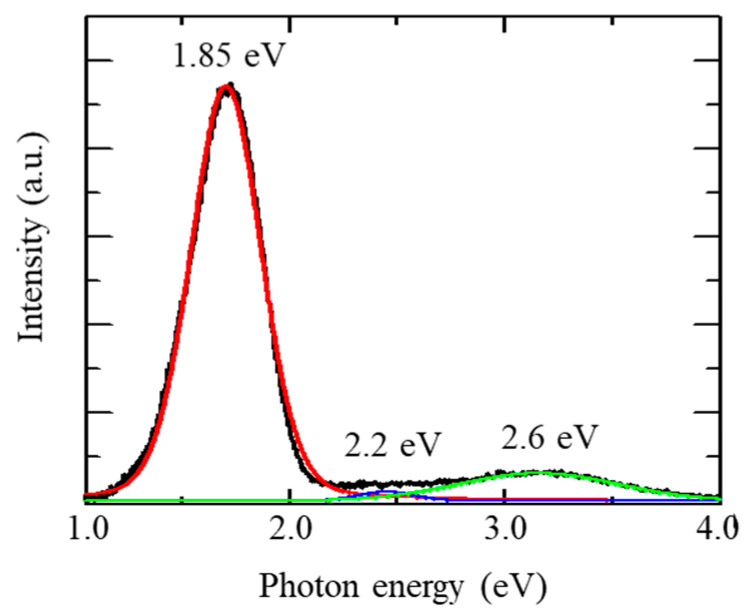
The obtained typical CL spectra from a thermally-oxidized SiO_2_ thin film. A typical CL spectrum from SiO_2_ (black line) and its decomposition into different contributions (red, green, and blue lines) by fitting Gaussian and Lorenz functions.

**Figure 4 materials-13-04490-f004:**
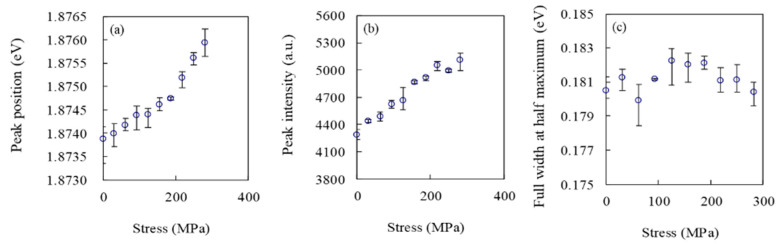
The relationships between applied tensile stress and CL spectral parameters; (**a**) peak position, (**b**) peak intensity, and (**c**) FWHM.

**Figure 5 materials-13-04490-f005:**
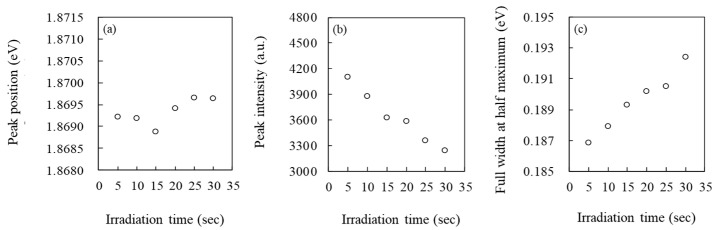
The relationships between EB irradiation time and CL spectral parameters; (**a**) peak position, (**b**) peak intensity, and (**c**) FWHM.

**Figure 6 materials-13-04490-f006:**
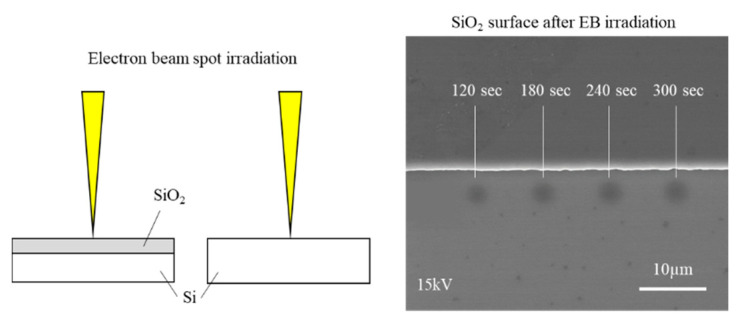
Schematic of EB irradiation along with the SEM image of the SiO_2_ surface. The four-circle pattern can be observed on the SiO_2_ surface after EB spot irradiation.

**Figure 7 materials-13-04490-f007:**
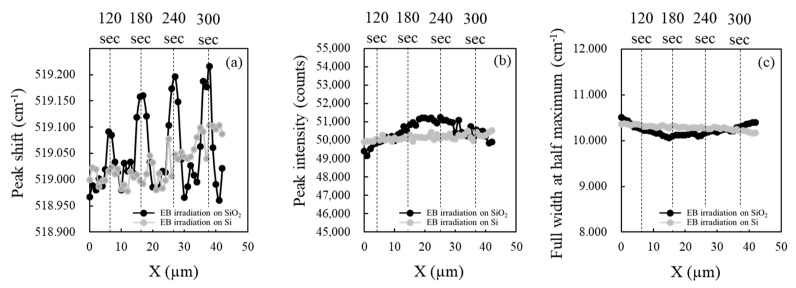
The influence of EB irradiation on the SiO_2_ and Si surface observed with Raman spectroscopy; (**a**) peak shift, (**b**) peak intensity, and (**c**) FWHM.

**Figure 8 materials-13-04490-f008:**
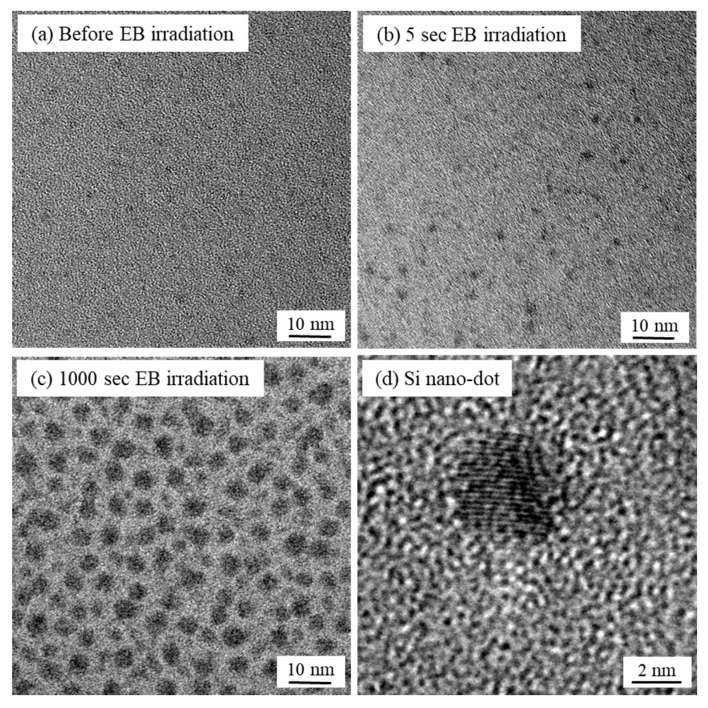
TEM images of SiO_2_ films after EB irradiation for (**a**) 0 s, (**b**) 5 s, (**c**) 1000 s, and (**d**) Enlarged view of an Si nanodot.

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
