# Peer review of "Cathodoluminescence Spectroscopic Stress Analysis for Silicon Oxide Film and Its Damage Evaluation"

_materials, 2020, doi:10.3390/ma13204490_

Round 1
Reviewer 1 Report
In my opinion this paper can be interesting to readers of Materials journal. English of the paper is good and meet the requirement of the journal.
I also find some mistakes for example:
- Abstract – please describe the scientificity of the test results obtained.
- Introduction – review is mainly based on Asian publications. For this reason, it is not possible to assess the validity and scientific novelty of the studies undertaken within paper with regard to world literature.
- Experimental Procedure – the names of the measuring equipment used should be given – model of equipment (manufacturer, city, country).
- Conclusions – chapter should be changed. There are no summary of all significant research results obtained by the Authors and written in the Results and Discussions
- Papers cited in the references (10 from all 15) are older then 10 years and 14 from all 15 are older then 5 years – Authors should include several modern papers from last 5 years.
- Minimum 12 papers from all 15 are wrote by authors from Asia –Japan, China and others. Please add some new (from the last 5 years) publications. Authors should include several modern papers (also from Europe and America).
The manuscript can be accepted for publication in Materials journal after MAJOR corrections.
Author Response
Response Paper:
I would like to thank all the reviewers for giving us fruitful comments and suggestions, which were very useful for revising the manuscript. The responses to all the questions are described as follows:
Reviewer 1
In my opinion this paper can be interesting to readers of Materials journal. English of the paper is good and meet the requirement of the journal. I also find some mistakes for example:
- Thank you very much for your comment.
Abstract – please describe the scientificity of the test results obtained.
- Thank you for the indication. The abstract has been revised entirely.
Introduction – review is mainly based on Asian publications. For this reason, it is not possible to assess the validity and scientific novelty of the studies undertaken within paper with regard to world literature.
- Thank you very much for the indication. As Reviewer 1 pointed out, several reference papers other than Asia have been newly added in the revised manuscript. 
Experimental Procedure – the names of the measuring equipment used should be given – model of equipment (manufacturer, city, country).
- Thank you again for the indication. The information on the model and manufacturer of all the equipments used here has been newly added in the revised manuscript.
Conclusions – chapter should be changed. There are no summary of all significant research results obtained by the Authors and written in the Results and Discussions
- Thank you again for the indication. The conclusion section has been entirely revised in the revised manuscript in consideration of the contents described in the results and discussions section.
Papers cited in the references (10 from all 15) are older than 10 years and 14 from all 15 are older than 5 years – Authors should include several modern papers from last 5 years. Minimum 12 papers from all 15 are wrote by authors from Asia –Japan, China and others. Please add some new (from the last 5 years) publications. Authors should include several modern papers (also from Europe and America).
- Thank you for the indication. As Reviewer 1 pointed out, many reference papers might have been very new. Several new papers from Europe and America have been added in the revised manuscript.
The manuscript can be accepted for publication in Materials journal after MAJOR corrections.
- Thank you very much for your decision. We hope that the revised manuscript is suitable for publication in MDPI Materials.

Reviewer 2 Report
This work presents the use of Cathodoluminescence (CL) spectroscopy for the quantitative stress analysis of thin films. The design is made of an SEM chamber that can directly measure CL. The nanocrystal formation during EB irradiation for CL stress analysis was verified using TEM and Raman.
Major revision: This work can be published if the following comments would be addressed.
Introduction:
Line 31- 34 Please state the reason for not using Raman for the large bandgap materials for the general readers.
Results and Discussion; Paragraph 1; please explain the other observed two peaks in Fig, 3.
According to Fig. 4 and 5, some parameters are not sensitive to EB irradiation time and are only sensitive to tensile as observed using CL spectroscopy stress analysis. At the same time, this work could draw a solid conclusion about the Raman stress due to the discrepancies of data between the Raman stress and CL stress analysis. Can this be related to the lack of full consideration and underestimation of data analysis of Fig. 4 and 5 and ignored insensitive parameters? The authors should speculate about this.
Author Response
Response Paper:
I would like to thank all the reviewers for giving us fruitful comments and suggestions, which were very useful for revising the manuscript. The responses to all the questions are described as follows:
Reviewer 2
This work presents the use of Cathodoluminescence (CL) spectroscopy for the quantitative stress analysis of thin films. The design is made of an SEM chamber that can directly measure CL. The nanocrystal formation during EB irradiation for CL stress analysis was verified using TEM and Raman.
Major revision: This work can be published if the following comments would be addressed.
Introduction: Introduction:
Line 31- 34 Please state the reason for not using Raman for the large bandgap materials for the general readers.
- In the case of a large bandgap material, the energy required when the electrons in the material are excited by the excitation light is large, and also the detected Raman scattering light becomes weak; therefore, the material is not suitable for Raman spectroscopy. The explanation has been newly added in the revised manuscript.
Results and Discussion; Paragraph 1; please explain the other observed two peaks in Fig, 3.
- Thank you for the indication. As Reviewer 2 pointed out, the explanation of the two peaks in Fig. 3 was lacking. The following description has been added in the revised manuscript.
- The red spectrum is attributed to the [ºSi-O-] band of the non-bridged oxygen hole center (NBOHC), the blue one is from [ºSi·] band of the trivalent Si, and the green one is from [ºSiº] band of the neutral oxygen vacancy defect.
According to Fig. 4 and 5, some parameters are not sensitive to EB irradiation time and are only sensitive to tensile as observed using CL spectroscopy stress analysis. At the same time, this work could draw a solid conclusion about the Raman stress due to the discrepancies of data between the Raman stress and CL stress analysis. Can this be related to the lack of full consideration and underestimation of data analysis of Fig. 4 and 5 and ignored insensitive parameters? The authors should speculate about this.
- Thank you for the indication. The purpose of this study is to find the possibility of CL as a stress analysis tool for SiO2 film. By applying external stress and EB irradiation, we newly found that only CL peak shift was a parameter sensitive to stress and insensitive to EB irradiation time. In addition, we tried to investigate the cause of a change of CL intensity and half bandwidth. By using Raman spectroscope, we found that EB irradiation produced Si nanocrystals into SiO2 film, which would have affected the change of CL intensity and half bandwidth with EB irradiation time. However, the mechanism cannot be concluded by only the data obtained until now. It is next step to conclude the influence of Si nanocrystals formation on CL spectral parameters. We would be very pleased if you could understand our situation.

Reviewer 3 Report
Dear Authors,
After revising the manuscript I would suggest improving the grammar point of the work. Otherwise I do not see a need for a major corrections.
Author Response
Response Paper:
I would like to thank all the reviewers for giving us fruitful comments and suggestions, which were very useful for revising the manuscript. The responses to all the questions are described as follows:
Reviewer 3
Dear Authors,
After revising the manuscript I would suggest improving the grammar point of the work. Otherwise I do not see a need for a major corrections.
- Thank you very much for the indication. We have polished the writing of the manuscript entirely.

Reviewer 4 Report
- The so called SiOx, in fact SiO and SiO2 is more common in practical process, and the stress behavior for them is also unknown. For this reason, please more clearly describe your research object SiO or SiO2 and the background of the research motivation.
- The stress change always along with the thermodynamic temperature for industrial application, so how to detect the change of temperature for this CL method? Such as how to realize and detect 1100 degree during your experimental process?
- More details should be provided for the increment rate calculation.
- The operating conditions of SEM need to be in vacuum, the thermal oxidation behavior of silicon matrix will change in thermodynamics and the formation of SiOx is accompanied by the generation and volatilization of bubbles at a vacuum of 1100 degrees. However, are these phenomena occurring during the experiment and how do you think about their influence to your results?
- The analysis results for products produced in silicon substrate surface under EB inradiation should also be better presented in this paper. For example, there is a lack of quantitative analysis of the results in Figure 8. Such as XRD, XPS for macro analysis should be carried out before TEM and SEM for microstructure analysis. This will make the work more practical.
- What is the effect of disproportionation between silicon and silicon oxide on the experimental results after the formation of silicon oxide? The relevant explanation or explanation should be given in discussion part.
- As a research paper, the author should use a few simple sentences to mention the direction of further research work that should be further optimized at the end of the article.
Author Response
Response Paper:
I would like to thank all the reviewers for giving us fruitful comments and suggestions, which were very useful for revising the manuscript. The responses to all the questions are described as follows:
Reviewer 4
- The so called SiOx, in fact SiO and SiO2 is more common in practical process, and the stress behavior for them is also unknown. For this reason, please more clearly describe your research object SiO or SiO2 and the background of the research motivation.
- Thank you for the indication. We made silicon oxide film by wet thermal oxidation of Si, but did not analyze the composition precisely. From the reason, we decided to use the expression, SiOx, for silicon oxide film. However, as Reviewer 4 pointed out, SiO2 is more common as the object prepared by wet thermal oxidation. Therefore, we have replaced SiOx to SiO2 throughout the revised manuscript. Also, the importance of stress analysis for SiO2 has been newly added in the revised manuscript.
- The stress change always along with the thermodynamic temperature for industrial application, so how to detect the change of temperature for this CL method? Such as how to realize and detect 1100 degree during your experimental process?
- Thank you for the interesting proposal. Yes, as Reviewer 4 mentioned, stress changes with temperature. In our experimental system, Joule’s heating might be served as a key technique for high temperature CL analysis under tensile stressing. Let us try to develop the test system in the near future.
- More details should be provided for the increment rate calculation.
- Thank you for the indication. We used a least-square method to calculate the photon energy increment rate to stress. As Reviewer 4 suggested, increment rate calculation for Fig. 4(a) has been newly added in the revised manuscript.
- The operating conditions of SEM need to be in vacuum, the thermal oxidation behavior of silicon matrix will change in thermodynamics and the formation of SiOx is accompanied by the generation and volatilization of bubbles at a vacuum of 1100 degrees. However, are these phenomena occurring during the experiment and how do you think about their influence to your results?
- As Reviewer 4 pointed out, the thermal oxidation behavior of silicon matrix will change in thermodynamics and the formation of SiOx is accompanied by the generation and volatilization of bubbles at 1100°C. In our study, all the CL analyses were conducted in a SEM at ambient temperature; that is, the specimen temperature was not controlled. In this study, by local irradiation of EB into a SiO2 surface, Si nanocrystals were formed. By the experimental fact, the local temperature at the irradiate point might have been high, but we guess that the temperature did not reach to 1100°C. Let us discuss and clarify that as our future work. We would like to explain the mechanism of Si nanocrystals formation by EB irradiation by using some kind of molecular dynamics simulations. Thank you very much for giving us very effective indication and for your understanding of our current situation and future work.
- The analysis results for products produced in silicon substrate surface under EB irradiation should also be better presented in this paper. For example, there is a lack of quantitative analysis of the results in Figure 8. Such as XRD, XPS for macro analysis should be carried out before TEM and SEM for microstructure analysis. This will make the work more practical.
- Thank you very much for the indication. Yes, as Reviewer 4 pointed out, more quantitative analysis and discussion are necessary to make the conclusion of our work stronger. XRD and XPS are effective and strong tools for this. However, we cannot address them soon because these facilities cannot be freely used for us. In the near future, we will do these experiments and will submit these results to another technical paper. We would be pleased if you could understand our situation and future plan.
- What is the effect of disproportionation between silicon and silicon oxide on the experimental results after the formation of silicon oxide? The relevant explanation or explanation should be given in discussion part.
- Yes, as Reviewer 4 pointed out, nonuniformity of silicon oxide might influence CL analyses. In this study, however, we conducted the experiments five times under the same condition, and the almost same trend was obtained throughout the experiments related to Fig. 4 and 5. Therefore, we think that the influence of the nonuniformity might exist, but it is probably small. The information on the number of the experiments has been newly included in the revised manuscript.
- As a research paper, the author should use a few simple sentences to mention the direction of further research work that should be further optimized at the end of the article.
- Thank you very much for the useful suggestion. Yes, to conclude the mechanisms of CL and Raman spectra changes by EB irradiation and stress application, further experiments and simulation should be made. The future direction of further experiments has been newly described in the conclusion section in the revised manuscript.

Round 2
Reviewer 1 Report
All my request were taken into account in the revised work. In my opinion, this paper should be accept and publish in MATERIALS Journal. Thank you for Authors.
Author Response
Dear Reviewer 1
Thank you very much for your decision. Your comments and suggestions were very useful for revising the manuscript. Thank you.
Reviewer 4 Report
The manuscript is acceptable for publication in the present form. It has been significantly improved and is now suitable for Materials.
All of the comments I made have been considered and there is a change/answer/comment to each.
I suggest acceptation after going through the manuscript more carefully for clarity, syntax and correctness. The English should be improved for the sake of clarity.
Author Response
Dear Reviewer 4
Thank you very much for your decision. Your comments and suggestions were very useful for revising the manuscript. We have checked and polished our writing. Thank you.